# The Vascular Involvement in Soft Tissue Fibrosis—Lessons Learned from Pathological Scarring

**DOI:** 10.3390/ijms21072542

**Published:** 2020-04-06

**Authors:** Chenyu Huang, Rei Ogawa

**Affiliations:** 1Department of Dermatology, School of Clinical Medicine, Tsinghua University, Beijing 100084, China; huangchenyu@mail.tsinghua.edu.cn; 2Department of Plastic, Reconstructive and Aesthetic Surgery, Nippon Medical School, Tokyo 113-8603, Japan

**Keywords:** soft tissue fibrosis, pathological scars, vascular involvement, endothelial dysfunction, endothelial-to-mesenchymal cell transition, pericytes, hepatic stellate cells, fibrocytes, myofibroblasts

## Abstract

Soft tissue fibrosis in important organs such as the heart, liver, lung, and kidney is a serious pathological process that is characterized by excessive connective tissue deposition. It is the result of chronic but progressive accumulation of fibroblasts and their production of extracellular matrix components such as collagens. Research on pathological scars, namely, hypertrophic scars and keloids, may provide important clues about the mechanisms that drive soft tissue fibrosis, in particular the vascular involvement. This is because these dermal fibrotic lesions bear all of the fibrotic characteristics seen in soft tissue fibrosis. Moreover, their location on the skin surface means they are readily observable and directly treatable and therefore more accessible to research. We will focus here on the roles that blood vessel-associated cells play in cutaneous scar pathology and assess from the literature whether these cells also contribute to other soft tissue fibroses. These cells include endothelial cells, which not only exhibit aberrant functions but also differentiate into mesenchymal cells in pathological scars. They also include pericytes, hepatic stellate cells, fibrocytes, and myofibroblasts. This article will review with broad strokes the roles that these cells play in the pathophysiology of different soft tissue fibroses. We hope that this brief but wide-ranging overview of the vascular involvement in fibrosis pathophysiology will aid research into the mechanisms underlying fibrosis and that this will eventually lead to the development of interventions that can prevent, reduce, or even reverse fibrosis formation and/or progression.

## 1. Introduction

Soft tissue fibrosis in important organs such as the heart, liver, lung, and kidney is a serious pathological process that is characterized by excessive connective tissue deposition. It is the result of chronic but progressive accumulation of fibroblasts and their production of extracellular matrix (ECM) components such as collagens. Soft tissue fibrosis decreases or obliterates organ function and can result in organ failure and death [1]. Indeed, it accounts for 45% of deaths in the United States [2]. However, the pathological mechanisms behind soft tissue fibrosis remain unclear.

Research on pathological scars, namely, hypertrophic scars and keloids, may provide important clues about the mechanisms that drive soft tissue fibrosis. This is because these dermal fibrotic lesions bear all of the fibrotic characteristics seen in soft tissue fibrosis. Moreover, their location on the skin surface means they are readily observable and directly treatable and therefore more accessible to research.

A particularly important focus in the field of cutaneous pathological scarring is the role of blood vessels: there is considerable evidence that suggests that these vessels and their dysfunctions contribute to the formation and progression of abnormally growing scars, as follows. First, pathological scars are characterized by chronic inflammation: this inflammation is particularly evident at the invasive edges of keloids and causes the erythematous and pruritic symptoms of these scars. This inflammation plays a key role in the development and progression of pathological scars, as demonstrated by the widely recognized beneficial effects of intralesional administration of anti-inflammatory drugs such as corticosteroid [3]. It is likely that this inflammation promotes excessive local angiogenesis, endothelial dysfunction, and vascular hyperpermeability that in turn facilitates the continued influx of inflammatory cells and factors, thereby setting up a vicious cycle of inflammation and abnormal vascular activities [4]. Second, several studies show a link between hypertension and pathological scarring. Thus, a cross-sectional study showed that keloid patients with systemic arterial hypertension have more severe keloids than keloid patients without hypertension [5]. Moreover, treatment with antihypertension pharmaceuticals such as angiotension-converting enzyme inhibitors (e.g., captopril and enalapril) can improve keloids [6]. Third, as shown by a comprehensive review of the studies in the field, there are multiple lines of evidence that suggest endothelial dysfunction plays a crucial role in scar pathophysiology [7].

To determine whether soft tissue fibrosis is also driven by abnormal vascular involvement, we will focus here on the cells and their functions that are known to participate in the abnormal vascular activity in cutaneous scar pathology. These cells include endothelial cells, which not only exhibit aberrant functions in pathological scars but also transdifferentiate into mesenchymal cells called myofibroblasts, which are highly contractile cells that produce large amounts of extracellular matrix (ECM). They also include cells that are in close communication with endothelial cells, namely, pericytes (including the highly specialized liver-specific pericytes called hepatic stellate cells) and fibrocytes: as we still show below, these cells can also convert into myofibroblasts in profibrotic conditions (Figure 1). This article will paint with broad strokes the roles that these cells play in the pathophysiology of different soft tissue fibroses. It should be noted that this is a very complex field that is still dogged by limited research effort and is thus rife with inconsistencies: therefore, this review focuses mainly on the papers that report high-quality, directly pertinent, and generally externally consistent data.

## 2. Endothelial Cells

The inner lining of blood vessels consists of monolayers of endothelial cells that are in direct contact with the cells and components of the blood. Consequently, these cells play a key role in vascular function, both as a barrier between the blood and the tissues and as an endocrine organ that actively regulates vascular dilation/constriction, immune responses, angiogenesis, and coagulation/fibrinolysis [8]. Endothelial cells contribute to fibrosis mainly in two ways: by becoming dysfunctional and by differentiating into highly fibrogenic mesenchymal cells called myofibroblasts.

### 2.1. Endothelial Dysfunction

Endothelial dysfunction refers to the impaired ability of endothelial cells to maintain vascular homeostasis [9,10]. This impairment arises partly from chronic exposure to low-level biomechanical stimuli such as shear, stretch, and hydrostatic pressure that eventually changes vascular tone, vascular permeability, and coagulation and vasomodulation activity. These stimuli and their effects on endothelial cells induce a vicious self-perpetuating cycle of inflammation, hypertension, microthrombosis, and endothelial cell hypersensitivity to these biomechanical stimuli [7].

#### 2.1.1. Endothelial Dysfunction in Skin Fibrosis

Endothelial dysfunction is a hallmark of skin fibrotic conditions such as pathological cutaneous scars and cystic fibrosis. For example, our logistic regression analysis showed that patients with keloids were significantly more likely than the non-keloid controls to have poor reactive hyperemia index and augmentation index values: both indices are measures of endothelial function [11]. In fact, endothelial dysfunction plays such an important role in pathological scar development and progression that these cutaneous scars can be classified as primary and secondary pathological scars depending on whether the associated endothelial dysfunction is due to congenital abnormalities (e.g., mutations) or to externally-generated systemic (e.g., hypertension) or local (e.g., the tension on the scar) conditions [4]. Similarly, in cystic fibrosis, two case-control studies on patients with and without this disease found that cystic fibrosis patients exhibited endothelial dysfunction in both the conduit and microvascular arteries, as shown by the brachial artery flow-mediated dilation test [12] and several measures of microvascular function [13]. This endothelial dysfunction appears to be contributed, at least in part, by oxidative stress since treating cystic fibrosis patients once with an oral antioxidant cocktail restored the endothelial function of the brachial artery and improved the systemic oxidative balance, as shown by lower lipid hydroperoxide (oxidative) level and the prevented reduction in α-tocopherol (antioxidant) level in the blood [14].

#### 2.1.2. Endothelial Dysfunction in Liver Fibrosis

There is also evidence that endothelial dysfunction participates in liver fibrosis: Graupera et al. showed that in carbon tetrachloride (CCl_4_)-induced cirrhotic rat livers, the intrahepatic microcirculation relaxes poorly when exposed to acetylcholine. They also elucidated some of the molecular mechanisms that underlie this effect: they found that cirrhosis associated with excessive levels of thromboxane A_2_ (TXA_2_, a mediator of liver fibrosis [15]) and that these high TXA_2_ levels in cirrhosis were due in turn to high activation of the enzyme that produces TXA2, namely, cyclooxygenase-1 [16]. These findings are supported by Rodríguez–Vilarrupla et al., who treated CCl_4_ cirrhotic rats with oral fenofibrate (25 mg/Kg/day) for seven days before sacrifice and liver analysis. Fenofibrate is an agonist of peroxisome proliferator-activated receptor (PPAR)α, which can be expressed by endothelial cells in the liver and downregulates COX-1. The fenofibrate treatment significantly improved the vasodilatory response of the liver to acetylcholine and reduced hepatic fibrosis. These beneficial effects associated with a drop in COX-1 and TXA_2_ production in vivo. Notably, Rodríguez–Vilarrupla et al. also showed that when hepatic endothelial cells from CCl_4_ cirrhotic rats were treated with fenofibrate in vitro (100 µM, 30 min), the treatment increased nitric oxide (NO) bioavailability [17]. In relation to this, Deng et al. found that when hepatic endothelial NO synthase phosphorylation and NO release was improved by gavage treatment of CCl_4_ cirrhotic rats with a soluble epoxide hydrolase inhibitor, these changes effectively improved hepatic endothelial dysfunction and liver fibrosis [18]. These findings together suggest that both low NO bioavailability and high production of TXA_2_ and COX-1 are responsible for the endothelial dysfunction in liver fibrosis.

Since we are discussing liver fibrosis here, we would like at this point to briefly digress to describe the highly specialized and differentiated endothelial cells that separate the blood cells in the sinusoidal lumen from the hepatocytes and hepatic stellate cells in the space of Disse [19]. These liver-sinusoid endothelial cells (LSECs) are characterized by non-diaphragmed fenestrae and the lack of a subendothelial basement membrane. These features allow this endothelium to serve as a dynamic filter [20]. Fibrosis in the space of Disse is preceded by loss of LSEC differentiation that leads to the defenestration of the LSEC layer and the development of a subendothelial basement membrane: this process is known as capillarization [21,22]. This capillarization associates with the endothelial dysfunction of the LSECs [23,24]. Initially, the defenestration can still be reversed by removing hepatotoxin [25]. However, once the basement membrane forms, the defenestration becomes irreversible [26]. Pasarin et al. showed with their rat model of non-alcoholic fatty liver disease that LSEC dysfunction precedes inflammation and early signs of fibrosis and is thus the earliest events in liver fibrogenesis [27]. As will be described in more detail below in the section on hepatic stellate cells, the loss of LSEC differentiation removes their ability to induce the quiescence of their hepatic stellate cell neighbors; the latter cells, once they are activated, are the source of the excessive ECM in liver fibrosis [27,28].

#### 2.1.3. Endothelial Dysfunction in Renal Fibrosis

Renal fibrosis also involves endothelial dysfunction. For example, Wu et al. showed that lipopolysaccharide-induced kidney injury in mice markedly declined the peritubular capillary perfusion. They also found that this response associated with rapid upregulation of intercellular adhesion molecule-1 and vascular adhesion molecule-1: this suggests that when renal tubule cells are stressed, they quickly induce vascular inflammation that promotes endothelial dysfunction [29]. In addition, the review of Molitoris et al. suggested that this inflammatory response in acute kidney disease may lead to congestion of the microvasculature; this congestion may act together with vasoconstriction and, later, with capillary rarefaction to induce hypoxia [30]. This hypoxia, in turn, promotes pro-fibrotic responses in tubule cells [31].

Lipphardt et al. also showed recently that endothelial dysfunction after unilateral ureteral obstruction is responsible for the ensuing kidney fibrosis. Interestingly, they found that this effect was mediated by increased kidney expression of Syndecan-4. This is a membrane-spanning proteoglycan that consists of a short cytoplasmic domain and an ectodomain that bears heparan sulfate chains [32]. It contributes to the scaffold of the endothelial glycocalyx (especially the glycocalyx of the glomerular endothelium), which is a dynamic network of membrane-bound proteoglycans and glycoproteins that covers the luminal side of endothelial cells [33]. The glycocalyx plays a key role in vascular physiology (and pathology), including in the kidney. Lipphardt et al. suggested that unilateral ureteral obstruction not only increases the expression of Syndecan-4, it also induces oxidative stress in the endothelial cells and that this may cause them to shed Syndecan-4 from the glycocalyx [32]. This, in turn, may induce interstitial fibrosis by promoting the impairment in the glycocalyx, which exposes adhesion molecules to the blood, thereby promoting leukocyte recruitment and inflammation [34]. Moreover, Lipphardt et al. suggested that Syndecan-4 shedding leads to the accumulation of its ectoderm in the tubulointerstitial matrix and that this attracts monocytes and induces renal fibroblasts to differentiate into fibrogenic myofibroblasts [32].

Another study that indicates the pivotal role of endothelial dysfunction in renal fibrosis is that of Ni et al. These authors examined the effect of FTY720 treatment on rats that had undergone 5/6 nephrectomy (a model of chronic kidney disease). FTY720 is an analog of sphingosine 1-phosphate that helps stabilize the endothelial barrier. When 5/6-nephrectomized rats underwent gavage with FTY720 a week after surgery, the treatment inhibited endothelial dysfunction in the renal microvasculature, thereby reducing renal fibrosis progression. These beneficial effects of FTY720 on renal endothelial function associated with improved endothelial NO bioavailability due to increased NO synthesis and increased expression of vascular endothelial growth factor (VEGF) and endothelial NO synthase, along with the reduced TGF-β1 expression [35].

#### 2.1.4. Endothelial Dysfunction in Cardiac Fibrosis

Balint et al. showed with a rat model that insular cortex ischemic stroke results in three histological features, namely, coronary microvascular endothelial dysfunction, myocardial inflammatory infiltration, and myocardial fibrosis. They also found that the fibrosis occurs downstream of the first two events [36]. Moreover, obesity-induced cardiovascular dysfunction associates with both endothelial dysfunction (excessive endothelium-dependent relaxation) and fibrosis, as indicated by increased cardiac expression of collagen 1α1, collagen 3α1, and periostin [37]. Huby et al. also showed that these effects are caused by excessive production of leptin by adipocytes [37]. And that obesity-mediated hyperaldosteronism can promote oxidative stress in endothelial cells and lead to their dysfunction [38].

### 2.2. Endothelial to Mesenchymal Cell Transition (EndoMT)

Myofibroblasts, as the name indicates, combine the functions of ECM synthesis in fibroblasts and contraction in smooth muscle cells. These functions allow them to regulate connective tissue remodeling. Myofibroblast contracture is usually long-lasting, unlike the short and rapid contraction of smooth muscle cells [39]. The hallmark of fibrosis is the chronic accumulation of myofibroblasts that not only produce excessive levels of ECM but also strongly contract that ECM, which stimulates further ECM production and thereby sets up a vicious circle of profibrotic activities [40]. Myofibroblasts develop from protomyofibroblasts, whose key features are the formation of contractile stress fibers. The subsequent acquisition of α-SMA expression indicates the development of differentiated myofibroblasts [41]. Consequently, α-SMA is the most widely applied molecular marker of myofibroblasts in the field [42].

Myofibroblasts in fibrotic diseases originate from several sources, including previously quiescent tissue fibroblasts, circulating bone marrow-derived CD34^+^ fibrocytes, and from the phenotypic transition of various cells, including endothelial cells. This phenotypic transition causes the loss of the original features of the cell and the new production of mesenchymal features [43]. These transitions are often regulated by transforming growth factor (TGF)-β [44]. In this section, we will focus on the myofibroblasts that evolve from endothelial cells via the transdifferentiation process called the endothelial-to-mesenchymal transition (EndoMT). EndoMT causes endothelial cells to lose their original adhesive properties, their apical-basal polarity, and their differentiated state, thereby becoming migratory, spindle-shaped, undifferentiated mesenchymal cells that readily invade adjacent tissues [45]. The acquisition of the mesenchymal phenotype is indicated by the new expression of mesenchymal molecules such as α-SMA and vimentin. Since the pathogenic role of EndoMT in soft tissue fibrosis is increasingly garnering interest in various fields [43], its role in organ fibrosis will be described here in detail.

As an aside, it should be noted that another source of myofibroblasts is epithelial cells: these cells transform into myofibroblasts via the well-known transdifferentiation process called the epithelial-mesenchymal transition (EMT). However, we will not discuss EMT-derived myofibroblasts further in this paper because recent cell fate-mapping studies in animal models show that EMT does not contribute directly to the pool of collagen-producing myofibroblasts during fibrosis in vivo [46,47,48].

#### 2.2.1. EndoMT-Derived Myofibroblasts in Skin Fibrosis

Lee et al. showed that five of twelve excised keloids exhibited early-stage EndoMT, as indicated by the expression of the mesenchymal marker vimentin by endothelial (CD31^+^) cells in the dermal vasculature. It was thought that the remaining seven keloids did not coexpress these markers because their endothelial cells had already gone through EndoMT. In the same study, Lee et al. showed that Wnt-3a, (whose signals play a critical role in fibrosis [49]), may drive EndoMT in keloids. Thus, they first showed that keloid tissues expressed high levels of Wnt-3a. They then observed that when human dermal microvascular endothelial cells were treated in vitro with Wnt-3a, their expression of endothelial cell markers (VE-cadherin and CD31) fell at the same time as their expression of mesenchymal cell markers (vimentin, slug, and α-SMA) increased. These changes coincided with morphological transformation of the endothelial cells into spindle-shaped cells. Thus, Wnt-3a-mediated EndoMT may contribute to the excessive dermal fibrosis that characterizes keloids [50].

#### 2.2.2. EndoMT-Derived Myofibroblasts in Renal Fibrosis

There is increasing evidence showing that EndoMT-derived myofibroblasts also play a key role in the tubulointerstitial fibrosis, tubular atrophy, and glomerulosclerosis that characterizes renal fibrosis [51]. For example, Zeisberg et al. showed with three animal models of chronic kidney disease (unilateral ureteral obstructive nephropathy, streptozotocin-induced diabetic nephropathy, and a model of Alport renal disease) that 30%–50% of the fibroblasts in the fibrotic murine kidneys simultaneously express both the endothelial marker CD31 and the mesenchymal markers fibroblast-specific protein-1 and α-SMA [52]. It was also shown that the kidneys in diabetic nephropathy bear α-SMA-positive myofibroblasts: these are located in the renal interstitium and glomeruli [53] and α-SMA expressions correlate inversely with renal function [54]. Similarly, Li et al. showed that compared to normal kidneys, the kidneys of mice with streptozotocin-induced diabetic nephropathy have significantly higher numbers and frequencies of endothelial-origin myofibroblasts. The early appearance of these cells after renal injury suggests that EndoMT participates in the initiation of renal interstitial fibrosis [55]. This notion was supported by evidence that EndoMT in streptozotocin-induced diabetic nephropathy associates with Smad3, which participates in the TGF-β1 pathway and plays an essential role in renal fibrosis [56]. Additionally, administering a specific Smad3 inhibitor early after streptozotocin treatment not only reduced renal fibrosis, it also abrogated EndoMT [57].

#### 2.2.3. EndoMT-Derived Myofibroblasts in Cardiac Fibrosis

Cardiac fibrosis occurs after myocardial infarction and the resulting death of cardiomyocytes. Since the mammalian heart has little capacity to regenerate, the dead cardiomyocytes are replaced with ECM, which in turn disrupts the normal myocardial structure and function [58]. Multiple lines of evidence suggest EndoMT contributes to cardiac fibrosis. For example, Widyantoro et al. showed that the cardiac fibrosis that associates with streptozotocin-induced diabetes mellitus in mice was characterized by EndoMT (as shown by the acquisition of mesenchymal markers by endothelial cells) as well as impaired cardiac microvascularization, cardiac fibrosis, and heart failure. They also found that cardiac EndoMT, as well as the other cardiac effects of diabetes, could be abolished by endothelial cell-specific knockout of endothelin-1 [59]. Moreover, Zeisberg et al. showed that 27–35% of fibroblasts (α-SMA^+^ or fibroblast-specific protein-1^+^) in murine myocardial fibrosis caused by pressure overload or chronic allograft rejection originate from endothelial cells. They also found that systemic administration of a recombinant human bone morphogenetic protein (BMP)-7 inhibited EndoMT and preserved the endothelial cell phenotype and thereby inhibited cardiac fibrosis progression [60].

#### 2.2.4. EndoMT-Derived Myofibroblasts in Other Soft Tissue Fibroses

There is some, albeit limited, evidence that EndoMT participates in lung and liver fibrosis as well. Thus, in the bleomycin-induced lung fibrosis mouse model, immunofluorescence analyses of lung showed significantly increased mesenchymal marker α-SMA and decreased endothelial marker VE-cadherin. Moreover, when the microRNA MiR-155 (which is known to participate in lung fibrosis) was specifically deleted in endothelial cells, the bleomycin-induced EndoMT was alleviated along with the lung fibrotic response [61]. Regarding liver fibrosis, patients and mice with liver cirrhosis bear a modest but still noticeable subpopulation of liver endothelial cells that express both CD31 and α-SMA. Moreover, when mice with liver cirrhosis were treated with BMP-7 (which prevents cultured murine liver endothelial cells from undergoing TGF-β1-induced EndoMT), their liver fibrosis improved [62].

## 3. Pericytes

Like endothelial cells, pericytes (also called mural cells or Rouget cells) can differentiate into myofibroblasts and play key roles in fibrogenesis. Pericytes are perivascular cells that are embedded in the microvascular basement membrane. They have rounded cell bodies that extend multiple long cytoplasmic processes along the length of the vessel, usually spanning several endothelial cells. The primary processes bear perpendicular secondary processes that partially encircle the vessel. Since these anatomical and morphological features can only be determined by electron microscopy, pericytes are generally identified on the basis of their general location and molecular markers such as nerve/glial antigen 2 (NG2, also known as chondroitin sulphate proteoglycan 4), platelet-derived growth factor (PDGF) receptor β (PDGFRβ), and desmin. However, it should be noted that a single pericyte-specific marker has not yet been identified. Pericytes communicate closely with their neighboring endothelial cells, either directly via specialized peg-like cytoplasmic fingers that insert into endothelial socket-like invaginations or indirectly via juxtacrine or paracrine signaling [63]. In terms of function, pericytes support the functional stability and structural integrity of the microvasculature by regulating vascular tone, blood flow [64], and capillary diameter [65], as well as by participating in basement membrane synthesis by secreting collagen IV, fibronectin, and laminin [66].

### 3.1. Pericytes in Skin Fibrosis

A transmission electron microscopy study in 1982 showed that half of the 22 hypertrophic scars and keloids that were examined contained pericytes bearing an inner peripheral band of microfilaments: such pericytes were never observed in normal skin or mature scar samples and thus were termed pericapillary myofibroblasts. It was thought that these myofibroblastic pericytes migrate from their perivascular location to the interstitial region, where they then become a source of profibrotic fibroblasts. The same study also observed that the vast majority of microvessels in these pathological scars had occluded lumens. This pointed to a chronically hypoxic environment (which is well-known to promote fibrosis [67]). It was suggested that this microvascular occlusion was caused in part by the proliferation of endothelial cells and in part by contraction of the pericapillary myofibroblastic cells [68]. Later, lineage tracing showed that pericytes are a major contributor to the myofibroblast population in dermal scarring: indeed, most of the collagen-producing α-SMA^+^ myofibroblasts in these scars originate from the PDGFRβ^+^ NG2^+^ perivascular subpopulation [69]. Moreover, stainings of excessive human scars for PDGFRβ, α-SMA, and a collagen synthesis marker (prolyl-4-hydroxylase β-subunit) indicated that pericytes that were migrating from the microvascular wall to perivascular space were adopting a mesenchymal phenotype and had started synthesizing collagen [70].

### 3.2. Pericytes in Kidney Fibrosis

Kidney fibrosis also involves activated pericytes that transdifferentiate into myofibroblasts, detach from the capillaries, migrate into the interstitium, and produce ECM. These activities directly induce fibrosis and indirectly contribute to capillary fragility and rarefaction and hypoxia in the kidney [71]. The lineage-tracing studies show that pericytes are one of the main sources of myofibroblasts in the interstitial kidney disease [72]. This is supported by a study using two murine models of kidney fibrosis (the unilateral ureteric obstruction model and the ischemia-reperfusion renal fibrosis model) crossed with recombinant mice in which the renal epithelial and stromal cells were genetically labeled: this study showed that during nephrogenesis, FoxD1^+^ mesenchymal cells become adult CD73^+^ PDGFRβ^+^ α-SMA^-^ pericytes, and that in fibrogenic conditions, these pericytes start expressing α-SMA. The study showed that the vast majority of the myofibroblasts in renal fibrosis were derived from the stromal precursors of pericytes. By contrast, EMT was not a source of myofibroblasts in kidney fibrosis [47].

The molecular mechanisms behind the transdifferentiation of pericytes into myofibroblasts in still being explored. However, multiple studies show that in kidney fibrosis, pericytes transdifferentiate into myofibroblasts via the PDGFR, VEGF, TGF-β, connective tissue growth factor, Wnt/LRP6, Ephrin B, and Hedghog pathways [73]. The identification of these pathways may aid further research and promote the development of pharmaceutical interventions.

Interestingly, numerous studies also suggest that renal pericytes may be the source of most of the body’s erythropoietin and that their transdifferentiation into myofibroblasts in kidney fibrosis impairs their eythropoietin production, thereby contributing to the anemia that characterizes advanced chronic kidney disease [73,74,75].

### 3.3. Pericytes in Lung Fibrosis

Hung et al. showed that after bleomycin-induced lung injury, FoxD1^+^ progenitor-derived pericytes expand and start expressing collagen I and α-SMA in fibrotic foci. It was concluded that pericytes are also the main source of myofibroblasts in pulmonary fibrosis [76]. By contrast, an earlier study found that while NG2^+^ PDFGRβ^+^ pericyte-like cells did proliferate in bleomycin-induced pulmonary fibrosis, they did not change into α-SMA^+^ myofibroblasts. The authors argued that this discrepancy relative to other studies finding this transition in fibrotic lungs and other organs may reflect organ-specific differences in terms of fibrotic responses and the use of different markers to identify pericyte subpopulations [48]. Whatever the reason, it seems that the role(s) pericytes play in pulmonary fibrosis remains to be determined.

## 4. Hepatic Stellate Cells (Abbreviated to HSCs in This Section)

The pericytes in the liver are called hepatic stellate cells (HSCs) (also called Ito cells and fat-storing cells). We have placed these cells in a section of their own due to their unique functions and interactions with liver sinusoid endothelial cells (LSECs): note that the LSECs have been described briefly in Section 2.1.2. HSCs are localized in the subendothelial space of the *Disse*, namely, between the antiluminal side of the LSECs and the basolateral surface of the hepatocytes. When in a quiescent state, HSCs are vitamin A-storing hepatic pericytes. However, when activated, they transdifferentiate into vitamin A-depleted myofibroblasts and secrete ECM, thereby forming scars in the space of Disse [77].

The close physical proximity of HSCs and LSECs allows these two cell types to communicate readily. Indeed, HSCs and LSECs depend on each other to maintain a quiescent and differentiated state, respectively. Once that balance has been upended, liver fibrosis ensues. The mechanisms by which LSECs maintain and promote HSC quiescence, and HSCs control LSEC differentiation, have been investigated. Thus, LSECs control HSC quiescence by suppressing HSC expression of α-SMA: confocal microscopy showed that when quiescent HSCs and differentiated LSECs from rats were co-cultured for three days, only 29.6% of the HSCs expressed α-SMA. However, when the quiescent HSCs were cultured by themselves, 70.5% acquired this marker (*p* < 0.01). Moreover, the ability of the LSECs to maintain HSC quiescence was lost when the LSECs had undergone capillarization: when previously quiescent HSCs were cultured with capillarized LSECs from cirrhotic rats, 81.2% of the HSCs started expressing α-SMA. A paracrine mechanism involving NO release by LSECs mediated the ability of LSECs to control HSC quiescence. Interestingly, HSCs that were activated by 3 days of homotypic culture reverted to a quiescent phenotype (namely, a compact cytoplasm bearing fat droplets and low α-SMA and collagen I expression) when they were subsequently cultured with freshly isolated LSECs for another three days (days 3–6) in the presence of exogenous VEGF. This is notable because, in the normal liver, hepatocytes and HSCs secrete VEGF. However, this paracrine production of VEGF drops markedly after capillarization and before cirrhosis. Thus, it is likely that VEGF stimulates the NO production of LSECs that maintains HSCs in a quiescent state. These observations together show that loss of liver expression of VEGF and LSEC capillarization may permit HSC activation and the resulting fibrosis [78].

As an aside here, it appears that LSEC-HSC interactions can also be modulated by biomechanical stimuli that are transmitted via the ECM. Liu et al. noted that early-stage liver fibrosis associates with pronounced sinusoidal angiogenesis while late-stage liver fibrosis associates with elevated collagen fiber accumulation. They speculated that early sinusoidal angiogenesis may provoke collagen condensation and ECM stiffening and that this, in turn, may cause LSEC capillarization and the subsequent HSC activation. These hypotheses were borne out by multiple experiments employing artificial fibrotic microniches that mimicked the hepatic sinusoids during the early and late stages of liver fibrosis [79].

The studies described above indicate that HSCs are activated to become fibrotic by capillarized LSECs. However, there is also evidence of the converse relationship that was mentioned above, namely, that HSCs can control LSEC differentiation: they are not merely passive effectors of fibrosis. First, several lines of evidence suggest that HSCs control LSEC differentiation indirectly via their production of collagens into the space of Disse. For examples, Ford et al. showed that when LSECs are monocultured on a collagen hydrogel with the same stiffness that is seen in normal liver sinusoids (6 kPa), they maintain well-defined fenestrae. By contrast, when they are cultured on stiff (36 kPa) collagen hydrogels that resemble the ECM produced by activated HSCs, they completely lose their fenestrae; they also start expressing CD31, which is normally not expressed by differentiated LSECs [80]. Second, there is evidence that HSCs can shape LSEC phenotype via a paracrine mechanism. This evidence relates to the fact that Kruppel-like factor 2 (KLF2) is naturally expressed early in cirrhosis to prevent progression of the disease. While this endogenous response is ultimately ineffective, augmenting KLF2 activity by simvastatin treatment can lessen the vascular dysfunction in cirrhosis. Marrone et al. showed that when cirrhotic (activated) HSCs from humans and rats were treated in vitro with simvastatin, their fibrotic phenotype improved: they expressed less α-SMA and procollagen I and exhibited less oxidative stress. This HSC-deactivating effect was also observed when HSCs were infected with KLF2-expressing adenovirus. When these simvastatin/adenovirus-treated (i.e., deactivated) HSCs were washed and then co-cultured with dedifferentiated LSECs, the phenotype of the LSECs improved: they showed elevated endothelial NO synthase and lower endothelin-1 expression. The paracrine communication from the deactivated simvastatin/adenovirus-treated HSCs was probably mediated by VEGF [81].

It should be noted that LSECs and HSCs also cooperate to maintain the homeostatic production/degradation of hepatic hyaluronic acid (HA), an ECM component: the HA in the liver is synthesized by HSCs and degraded by LSECs. This homeostatic relationship is disrupted in liver fibrosis and results in increasingly higher HA levels in the serum as liver fibrosis progresses [82].

## 5. Fibrocytes

Fibrocytes (also known as perivascular fibroblasts or adventitial cells) differ from pericytes in that they surround arterioles and have no physical connection with endothelial cells [83]. The term ‘fibrocyte’ [84] comes from the words ‘fibroblast’ and ‘leukocyte’, which reflects the fact that they are a small subpopulation of circulating leukocytes (0.1%–0.5% of all nonerythrocytic cells in peripheral blood [85]) that resemble fibroblasts in terms of morphology and their secretion of ECM components (e.g., collagen I [86]); notably, since these cells are derived from monocytes, they also express hematopoietic markers, namely, CD34, CD43, CD45, lymphocyte-specific protein (LSP)-1, and major histocompatibility complex class II. Other fibrocyte markers are collagen I and III [87] and the chemokine receptor CXCR4 [88].

### 5.1. Fibrocytes in Skin Fibrosis

On cutaneous skin injury, peripheral blood fibrocytes migrate into the wound (together with other circulating inflammatory cells) and differentiate into fibroblasts/myofibroblasts [89]. These fibrocyte-derived fibroblasts then produce ECM. This is shown by studies on biomaterial implants, which inevitably trigger the recruitment of fibrocytes and their production of a fibrotic capsule around the implant: Thevenot et al. showed that the fibrocyte numbers that are recruited to an implant correlate directly with the collagen production at the implant interface [90].

Several lines of evidence suggest that fibrocyte-derived fibroblasts in the skin can also differentiate further into fibrotic myofibroblasts that express α-SMA and promote pathological scarring. For example, keloid-derived fibroblasts bear fibrocyte markers (CD34 and CD86), which suggests that these pathological fibroblasts could be derived from fibrocytes [91]. It should be noted that like the other myofibroblast-forming cells described above, the differentiation of fibrocytes into α-SMA-expressing contractile myofibroblasts is driven by TGF-β [92]. This is supported by the fact that when implants that excite fibrocyte-mediated fibrosis release a TGF-β inhibitor (SB431542), fibrocyte-derived myofibroblast differentiation is suppressed [93]. (As an aside, fibrocytes can also differentiate into adipocytes: when they are treated with the PPARγ agonist troglitazone, they stain positively with Oil Red O due to their lipid accumulation. Interestingly, treatment of fibrocytes with troglitazone antagonizes the ability of TGFβ to induce fibrocytes to express α-SMA mRNA and protein [92]. A similar phenomenon was observed in vivo in mice that were implanted with a biomaterial implant. While this implant normally provoked fibrocyte-mediated fibrosis, attachment of a mini-osmotic pump to the implant that delivered an adipogenic cocktail reduced fibrocyte-mediated fibrosis. Thus, targeting the differentiation potential of fibrocytes could help to alter the fibrotic tissue responses to biomaterial implants, as well as prevent pathological scarring [93].).

There are also other lines of evidence that suggest fibrocytes participate in pathological scarring. Thus, Travis et al. showed that in a porcine hypertrophic scar model, fibrocytes (LSP-1^+^ CD45^+^ procollagen-1^+^) arrived in the wound/scar in two waves, namely, 2–4 days after wounding and later during scar formation and remodeling [94]. Moreover, post-burn hypertrophic scars have greater numbers of fibrocytes (LSP-1^+^ N-terminal propeptide of type I collagen^+^) than mature scars (2.4 vs. 1.4/high power field, *p* < 0.05). By contrast, normal skin lacks fibrocytes [95]. Similarly, multicolor fluorescence-activated cell analysis showed that keloid samples have more fibroctyes (CD45RO^+^ 25F9^+^ MRP8/14^+^) than normal scars [96].

### 5.2. Fibrocytes in Lung Fibrosis

Fibrocytes also contribute to the pathogenesis of pulmonary fibrosis. Immunofluorescence and confocal microscopy analyses show that the lungs of patients with idiopathic pulmonary fibrosis bear fibrocytes [97]. Moreover, in a murine model of bleomycin-induced pulmonary fibrosis, the fibrocytes (CD45^+^ Collagen I^+^ CXCR4^+^) infiltrated as early as two days after bleomycin exposure. This is earlier than when significant collagen deposition is first detected (between 4 and 20 days). The fibrocyte numbers in the lungs peaked on day 8 and started to decrease by the last two observation time points (days 16 and 20) [88]. This decrease may reflect the change in phenotype of the fibrocytes into myofibroblasts: this notion was supported by in vitro observations [88,98]. Notably, when mice with bleomycin-induced pulmonary fibrosis were given 16 daily systemic injections of an antibody specific for CXCL12 (the chemokine that binds to the fibrocyte marker CXCR4), fibrocyte recruitment to the lungs was significantly attenuated along with collagen deposition and lung fibrosis [88]. Similarly, when mice with bleomycin-induced pulmonary fibrosis were orally treated every day with the anti-fibrotic agent pirfenidone (300 mg/Kg/day) for two weeks starting 10 days after bleomycin treatment, the fibrocyte (CD45^+^ Collagen I^+^) numbers in the lung dropped significantly on day 14 (*p* = 0.0097); this drop was mirrored by a fall in collagen content on day 28 (*p* = 0.0012) [99].

### 5.3. Fibrocytes in Other Soft Tissue Fibroses

Fibrocytes also participate in liver, cardiac, and renal fibrosis. In terms of liver fibrosis, fibrocytes constitute 5%–6% of all collagen-expressing cells in livers that have been injured by bile-duct ligation or CCl_4_ [100,101]. Moreover, when fibrocytes are specifically depleted in the thioacetamide-induced liver fibrosis mouse model by employing a herpes simplex virus thymidine kinase/valganciclovir suicide gene strategy, fibrillar collagen deposition and liver-specific cellular damage dropped significantly [102]. By contrast, a similar study that used clodronate to deplete monocytes and monocyte-derived cells, including fibrocytes, found that this depletion did not attenuate the progression of CCl_4_-induced liver fibrosis in mice [103]. However, this discrepant finding may reflect the broad depletion of monocyte-lineage cells. Further studies on the role of fibrocytes in liver fibrosis are warranted.

In terms of cardiac fibrosis, Sopel et al. showed that when myocardial fibrosis is induced in rats by angiotensin II infusion, fibrocyte progenitor cells are quickly recruited to the myocardium and differentiate into myofibroblasts; these events occur before collagen deposition starts [104]. The study of Xu et al. with CCR2-knockout mice then showed that the recruitment of the bone marrow-derived fibroblast precursor cells of fibrocytes to the myocardium of mice infused with angiotensin II is mediated by CCR2: unlike the wild-type mice, the CCR2-knockout mice demonstrated little fibrosis and few fibrocytes in the myocardium [105].

In terms of kidney fibrosis, Sakai et al. showed that renal fibrosis involves angiotensin II and its two receptors, AT1R (whose blockade reduces the risk of renal disease) and AT2R (which appears to counter the effects of AT1R and thus is organoprotective). First, the renal fibrosis caused by unilateral ureteral obstruction was reduced by valsartan-mediated blockade of AT1R and worsened by AT2R-knockout. The improvement caused by valsartan was mirrored by decreased fibrocyte infiltration and collagen deposition in the kidney. The opposite was observed when AT2R was knocked out. These findings were confirmed in another model of renal fibrosis, namely, the chronic angiotensin II-infusion model. The study suggested that AT1R/AT2R signaling shapes both the production of fibrocytes in the bone marrow and their fibrogenic activation [106].

## 6. Interactions between Myofibroblasts and Endothelial Cells

Myofibroblasts are pathogenic to endothelial cells. This can be seen in patients with idiopathic pulmonary arterial hypertension (PAH). Thus, Xu et al. first showed that the pulmonary artery endothelial cells in PAH exhibit decreased mitochondrial function [107]. The in vitro study of Sakao et al. then found that when human pulmonary microvascular endothelial cells were cultured with myofibroblast-like cells, their expression of a mitochondrial marker (mitotracker red) became disrupted and rarefied. Moreover, their expression of superoxide dismutase-2 dropped. By contrast, when the microvascular endothelial cells were co-cultured with pulmonary arterial fibroblast-like cells, they maintained their normal filamentous mitochondrial reticulum [108].

Conversely, endothelial cells can inhibit the formation of myofibroblasts. Thus, Yang et al. showed that when mice with unilateral ureteral obstruction-induced renal fibrosis were injected via the tail vein with putative endothelial progenitor cells (pEPCs), the mice were protected against renal fibrosis. This associated with significantly reduced expression of α-SMA and collagen IV in the kidneys. The pEPCs appeared to act by inhibiting pericyte detachment from endothelial cells and the pericyte-myofibroblast transition [109].

Endothelial cells can also promote the healthy differentiation of fibroblasts into myofibroblasts: this was shown indirectly by the study of Miyazaki et al. on wound healing. This study used a dermal wound healing mouse model and found that calpastatin, an endogenous inhibitor of calpains, was enriched in pre-existing blood vessels in the wound but not in newly formed capillaries. When endothelial cells were transformed genetically to overexpress calpastatin, dermal wound healing was delayed. This effect associated with poor myofibroblast accumulation in vivo, which reduced the appropriate fibrogenic responses in the wound bed and slowed healing [110]. Nieves Torres et al. also found that endothelial cells can regulate fibroblastic differentiation into myofibroblasts when they studied the venous neo-intimal hyperplasia that associates with hemodialysis vascular access. The hallmark of this condition is hypoxia that causes fibroblasts to differentiate into α-SMA-expressing myofibroblasts. Nieves Torres et al. showed by in vitro experiments that when blood outgrowth endothelial cells (BOEC, a subset of circulating progenitor cells with an endothelial cell phenotype), were in contact with fibroblasts, they decreased the conversion of the fibroblasts into myofibroblasts. This associated with reduced fibroblast migration and size and down-regulation of fibroblast expression of pro-angiogenic genes, including VEGFA, PDGF, fibroblast growth factor-1, and matrix metalloproteinase-2 [111].

## 7. Summary

This paper briefly and with broad strokes describes what is known about the vascular involvement in soft tissue fibrosis. Based on the lessons learned from pathological scar research, it focuses on the vascular cells that contribute directly in one way or another to fibrosis, namely, endothelial cells, pericytes (and their hepatic equivalent, hepatic stellate cells), and fibrocytes. The studies reviewed in this paper suggest that these vascular cells not only play important roles in soft tissue fibroses, they also act in broadly similar ways: in particular, they can convert into heavily profibrotic myofibroblasts in response to the fibrotic stimulus. These studies also suggest that the spatial and structural proximity of the different vascular cells appear to potentiate their dynamic interactions and integral functional changes during fibrotic progression.

There are many questions that remain about vascular cell-mediated fibrosis. In particular, despite the broad similarities between the vascular cells in terms of their roles in fibrosis, these cells have distinctive functions and structures in specific organs: examples of this are the renal pericytes, which are the source of most of the body’s erythropoietin, and the LSECs, which are the only endothelial cells in the body that have non-diaphragmed fenestrae and no basement membrane. Further studies on how these organ-specific features shape the fibrotic process and destination and its deleterious effects are warranted. Another important question is the role of endothelial cell autophagy in fibrosis. Autophagy is the self-degradative process (e.g., clearing damaged organelles) [112]. There is some limited evidence that autophagy in endothelial cells can shape fibrosis. For example, Ruart et al. showed that autophagy is upregulated in LSECs after CCl_4_ treatment and protects these cells from oxidative stress: however, this defensive autophagic response is ultimately overcome and endothelial cell dedifferentiation and the resulting fibrogenic activation of hepatic stellate cells ensues [113]. Conversely, a recent study with murine models of central nervous system demyelinating diseases showed that endothelial cell autophagy can have pro-fibrotic effects: soon after a demyelinating injury to the myelin sheath in the murine brain, brain microvascular endothelial cells autophage IgG-opsonized myelin debris. This has pathological effects because this autophagic response induces EndoMT and fibrosis [114]. Thus, autophagy in endothelial cells may play both protective and pathological roles in profibrotic conditions. Another interesting question relates to the endothelial glycocalyx, which we mentioned in Section 2.1.3. This structure lines the luminal side of vascular endothelial cells and is a critical signaling platform that integrates mechanical forces and biochemical signaling [115]. Section 2.1.3 shows that oxidative stress in renal endothelial cells causes loss of the endothelial glycocalyx and that this promotes kidney fibrosis [32]. However, there is some limited evidence that organs differ in their endothelial glycocalyx morphology and that these morphologies are in turn altered in specific ways by different pathogenic stimuli [116]. Comparative investigations on this structure in different organs with disparate external stimuli may further help to elucidate the fibrotic mechanisms in specific organs.

We hope that this brief but wide-ranging overview of the vascular involvement in fibrosis pathophysiology will aid research into the mechanisms underlying fibrosis and that this will eventually lead to the development of interventions that can prevent, reduce, or even reverse fibrosis formation and/or progression.

## Figures and Tables

**Figure 1 ijms-21-02542-f001:**
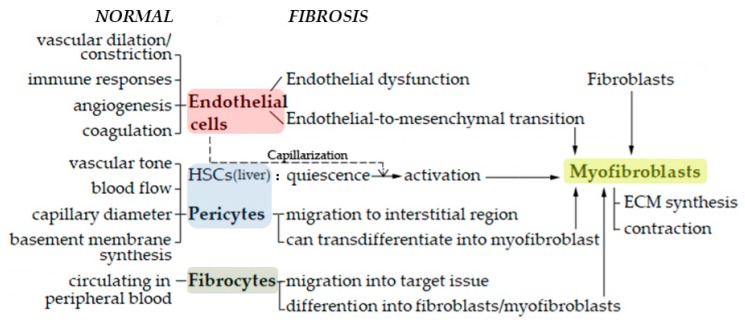
A brief outline of the participation of endothelial cells, pericytes, hepatic stellate cells (HSCs), fibrocytes and myofibroblasts in soft tissue fibrosis, in comparison to their normal roles.

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
