# Peer review of "The Vascular Involvement in Soft Tissue Fibrosis—Lessons Learned from Pathological Scarring"

_ijms, 2020, doi:10.3390/ijms21072542_

Round 1

Reviewer 1 Report

Dr. Huang and Dr. Ogawa reviewed important issue of soft tissue fibrosis with the view of vascular pathological involvement. Reviewer has made several comments below.

1. Since fibrosis is seen in most of organs, some organs are missing in the review (although the final goal of the review was to focus on soft tissue). In section 2. Endothelial cells, organs such as lung and heart are missing, lung and liver is missing in section 2.5 EndoMT, liver, lung, and heart in section 3 pericytes, and finally liver, heart, and kidney is missing in section 5. Fibrocytes. Review should be comprehensive due to the nature of the publication.

2. Schematic picture or table, showing which cellular properties or events are important in each organ and to show the connection between each organ to soft tissue fibrosis, are needed to contribute to the reader’s understanding of the current issue (eg Br J Dermatol. 2017 Nov;177(5):1248-1255.).

3. I recommend to add a paragraph showing the similarities and differences of fibrosing patterns between soft tissue and each organ.

Author Response

Dr. Huang and Dr. Ogawa reviewed important issue of soft tissue fibrosis with the view of vascular pathological involvement. Reviewer has made several comments below.

  1. Since fibrosis is seen in most of organs, some organs are missing in the review (although the final goal of the review was to focus on soft tissue). In section 2. Endothelial cells, organs such as lung and heart are missing, lung and liver is missing in section 2.5 EndoMT, liver, lung, and heart in section 3 pericytes, and finally liver, heart, and kidney is missing in section 5. Fibrocytes. Review should be comprehensive due to the nature of the publication.

→ Thank you for this point. To address it, we have added sections on nearly all of the missing organs to the paper i.e. heart was added to the Endothelial Dysfunction section; liver, lung, and heart were added to the EndoMT section; and liver, heart, and kidney were added to the Fibrocyte section. The Pericyte section already discussed lung data and the HSC section is actually a section on pericytes in the liver since HSCs are liver-specific pericytes. We have now made the latter point clear in the paper and apologize for the confusion. It was not possible to add lung data to the Endothelial Dysfunction section or heart data to the Pericyte section because we focused our review on papers that reported high-quality, directly pertinent research whose results were largely consistent with those described by similar papers: the data on these missing organs are poor and therefore we excluded them from the paper.

  1. Schematic picture or table, showing which cellular properties or events are important in each organ and to show the connection between each organ to soft tissue fibrosis, are needed to contribute to the reader’s understanding of the current issue (eg Br J Dermatol. 2017 Nov;177(5):1248-1255.).

→ Thank you for this helpful suggestion. We have added a figure to demonstrate the key roles of endothelial cells, pericytes, HSCs, and fibrocytes in the normal state and that they all convert to myofibroblasts in profibrotic conditions. It should be noted that the literature is still not advanced enough to draw other parallels between fibroses in different organs. This point, and the need for organ comparative studies, has been emphasized in the penultimate paragraph of the Summary.

  1. I recommend to add a paragraph showing the similarities and differences of fibrosing patterns between soft tissue and each organ.

→ Again, the literature is still not sufficiently advanced to determine the similarities and differences between the soft tissue and each organ in terms of fibrosing patterns.

Reviewer 2 Report

Dear authors,

After reviewing your manuscript, I have some suggestion for improving this manuscript:

  1. This manuscript is not easy to read, especially for a newbie in this field. The main problem I think logic is not telling in simple. If the author could modify the description to more directly, it would be easier to read.
  2. Description is not totally comprehensive. As I search several keywords such as "endothelial dysfunction" and "liver fibrosis", I found some interest references not cited in this manuscript. e.g., Ruart et al. describe the oxidative-stress-induced endothelial autophagy might contribute to liver fibrosis in acute liver injury (10.1016/j.jhep.2018.10.015). I suggest the authors could survey the references more comprehensively.
  3. Figures or schemes needed. This manuscript is focused on the role of vascular impairment in soft tissue fibrosis. The detail mechanism is complex and not easy to understand by words. If applicable, some schemes for summarize the description would be perfect.
  4. References style need to be uniform.
  5. Perspective needed. In the conclusion section, the authors just summarize the recent finding in vascular involvement in soft tissue fibrosis. For the readers, we want to see more about your suggestion in unsolved puzzles and point the sight. If applicable, please reinforce this section more.

Author Response

After reviewing your manuscript, I have some suggestion for improving this manuscript:

  1. This manuscript is not easy to read, especially for a newbie in this field. The main problem I think logic is not telling in simple. If the author could modify the description to more directly, it would be easier to read.

→ Thank you for your comment. To address this point, we have striven to improve the structure of the paper and its flow of concepts. We acknowledge that despite this, the paper continues to be quite complex. This reflects the vastness of the fields being covered, the incomplete and occasionally contradictory nature of the research data in these fields, and the sheer complexity of the cellular and molecular interactions that promote (or prevent) fibrosis. Since the review is intended to provide a broad but comprehensive overview of the vascular involvement in fibrosis, it has been necessary to sharply compress the details; this has increased the density of the information in the paper, which in turn has decreased the reading comfort. We apologize sincerely for this and hope that the changes we have made have made the paper easier to read.

  1. Description is not totally comprehensive. As I search several keywords such as "endothelial dysfunction" and "liver fibrosis", I found some interest references not cited in this manuscript. e.g., Ruart et al. describe the oxidative-stress-induced endothelial autophagy might contribute to liver fibrosis in acute liver injury (10.1016/j.jhep.2018.10.015). I suggest the authors could survey the references more comprehensively.

→ Our review focuses on publications that report high-quality and directly pertinent data that are generally consistent with the findings of other papers. It is certainly true that there are many elements in the field that we have not addressed, many of which are very intriguing and deserve further attention. However, their omission from this review reflects the fact that the research in these areas is still tentative, limited, indirect, and/or contradictory. The role of autophagy in fibrosis is in fact a good example of this: the literature on fibrosis-related autophagy in endothelial cells, fibroblasts, epithelial cells, and hepatic stellate cells is limited to isolated references (i.e. 10.1016/j.jhep.2018.10.015, doi: 10.4049/jimmunol.1801515, doi: 10.1002/jcp.29187, and doi: 10.1177/0960327119891212) that somewhat contradict each other. The inclusion of these patchy observations would, we fear, further weaken the readability of our paper. However, we do understand your point and have addressed it by adding a paragraph to the end of the article that mentions autophagy (including the paper of Ruart et al.) and indicates that there are many elements in the field that warrant much more research.

  1. Figures or schemes needed. This manuscript is focused on the role of vascular impairment in soft tissue fibrosis. The detail mechanism is complex and not easy to understand by words. If applicable, some schemes for summarize the description would be perfect.

→ Thank you for this helpful suggestion. We have added a figure to demonstrate the key roles of endothelial cells, pericytes, HSCs, and fibrocytes in the normal state and that they all convert to myofibroblasts in profibrotic conditions. We hope that the figure will aid the readability of our paper.

  1. References style need to be uniform.

→ Thank you, we have ensured that the references are cited in a uniform manner.

  1. Perspective needed. In the conclusion section, the authors just summarize the recent finding in vascular involvement in soft tissue fibrosis. For the readers, we want to see more about your suggestion in unsolved puzzles and point the sight. If applicable, please reinforce this section more.

→ Thank you for this important comment. To address it, we added a paragraph to the Summary that suggests areas in fibrosis that deserve further exploration, including comparisons of organs in terms of type of vascular involvement and fibrosis, the role of autophagy in vascular cells, and how the endothelial glycocalyx varies depending on the organ and type of profibrotic insult.

Reviewer 3 Report

This review outlines the cellular players in fibrosis and scar formation in several organ systems.  It gives a good overview of the types of scarring that occur and the sequence in which they occur in.  I think that the manuscript may be improved with the following suggestions:

The authors need to make a clear distinction of EMT and EndoMT.  I think the terms are getting confused in the field so there should be a clear definition of both to avoid making more confusion of these terms.

Reference 2 does not seem to have any facts that fibrosis accounts for 45% of all deaths in the US and should be deleted.

Section 2.5, 2nd paragraph, last sentence:  reword sentence.

Section 2.6:  need to define HDMECs, HSs and the function of Wnt-3a.

Section 2.8 needs to be developed.  Please give more details on cardiac fibrosis and what parts of the heart are prone in its development.

Section 3.2:  2nd paragraph is just one sentence without development and context.

Format of the references is inconsistent. 

Author Response

This review outlines the cellular players in fibrosis and scar formation in several organ systems.  It gives a good overview of the types of scarring that occur and the sequence in which they occur in.  I think that the manuscript may be improved with the following suggestions:

  1. The authors need to make a clear distinction of EMT and EndoMT.  I think the terms are getting confused in the field so there should be a clear definition of both to avoid making more confusion of these terms.

→Thank you for this comment. We agree that EMT and EndoMT are becoming confused and have made some changes to our paper so that the reader clearly understands that we are focusing on EndoMT, since lineage-tracing studies show that EMT does not participate in fibrosis.

  1. Reference 2 does not seem to have any facts that fibrosis accounts for 45% of all deaths in the US and should be deleted.

→ Thank you very much for reading our paper so carefully. Reference 2 states in the first two lines of a boxed insert (BOX1) on page 585 that “The United States government estimates that 45% of deaths in the United States can be attributed to fibrotic disorders”. We have therefore retained this reference.

  1. Section 2.5, 2nd paragraph, last sentence:  reword sentence.

→ Thank you, we have reworded that sentence.

  1. Section 2.6:  need to define HDMECs, HSs and the function of Wnt-3a.

→ Thank you, we have eliminated the abbreviations HDMEC and HS. We also stated that Wnt-3a signaling is known to play a critical role in fibrosis and that it is upregulated in keloids.

  1. Section 2.8 needs to be developed.  Please give more details on cardiac fibrosis and what parts of the heart are prone in its development.

→ Thank you. We altered this section to state that “Cardiac fibrosis occurs after myocardial infarction and the resulting death of cardiomyocytes. Since the mammalian heart has little capacity to regenerate, the dead cardiomyocytes are replaced with ECM, which in turn disrupts the normal myocardial structure and function [58]”. Ref 58 is doi: 10.1007/s00018-013-1349-6.

  1. Section 3.2:  2nd paragraph is just one sentence without development and context.

→ Thank you. That paragraph aimed to showcase novel molecular mechanisms for the transdifferentiation of pericytes into myofibroblast in kidney fibrosis. While this field is still in its infancy, the potential of this field to identify and promote the development of drug targets is large. This point has been added to that paragraph.

  1. Format of the references is inconsistent. 

→ Thank you, we have ensured that the reference style is now consistent.

We are deeply grateful to the editor and the reviewers for reading our manuscript so carefully and for providing important and constructive comments and suggestions. We believe that the revised paper addresses all of their concerns and that the paper has been substantially improved.

Round 2

Reviewer 1 Report

The revised manuscript will be read by readers with interest.